# Acute Intravesical Capsaicin for the Study of TRPV1 in the Lower Urinary Tract: Clinical Relevance and Potential for Innovation

**DOI:** 10.3390/medsci10030050

**Published:** 2022-09-10

**Authors:** Karl-Erik Andersson, Delphine Behr-Roussel, Pierre Denys, Francois Giuliano

**Affiliations:** 1Institute for Regenerative Medicine, Wake Forest University School of Medicine, Winston Salem, NC 27101, USA; 2Division of Clinical Chemistry and Pharmacology, Lund University, 22242 Lund, Sweden; 3Pelvipharm SAS, 78180 Montigny-le-Bretonneux, France; 4Neuro-Uro-Andrology R.Poincare Academic Hospital, AP-HP, 104 bvd R. Poincare, 92380 Garches, France; 5Faculty of Medicine, U1179 Inserm/Versailles Saint Quentin University, Paris Saclay, 78180 Montigny-le-Bretonneux, France

**Keywords:** experimental pharmacology, overactive bladder, neurogenic detrusor overactivity, interstitial cystitis, pain bladder syndrome, bladder outlet obstruction

## Abstract

Capsaicin acts on sensory nerves via vanilloid receptors. TRPV1 has been extensively studied with respect to functional lower urinary tract (LUT) conditions in rodents and humans. We aimed to (1) provide background information on capsaicin and TRPV1 and its mechanisms of action and basis for clinical use, (2) review the use of acute intravesical capsaicin instillation (AICI) in rodents to mimic various LUT disorders in which capsaicin sensitive C-fibers are involved and (3) discuss future innovative treatments. A comprehensive search of the major literature databases until June 2022 was conducted. Both capsaicin-sensitive and resistant unmyelinated bladder afferent C-fibers are involved in non-neurogenic overactive bladder/detrusor overactivity (OAB/DO). AICI is a suitable model to study afferent hyperactivity mimicking human OAB. Capsaicin-sensitive C-fibers are also involved in neurogenic DO (NDO) and potential targets for NDO treatment. AICI has been successfully tested for NDO treatment in humans. Capsaicin-sensitive bladder afferents are targets for NDO treatment. TRPV1-immunoreactive nerve fibers are involved in the pathogenesis of interstitial cystitis/painful bladder syndrome (IC/PBS). The AICI experimental model appears relevant for the preclinical study of treatments targeting bladder afferents for refractory IC/BPS. The activity of capsaicin-sensitive bladder afferents is increased in experimental bladder outlet obstruction (BOO). The AICI model may also be relevant for bladder disorders resulting from C-fiber hyperexcitabilities related to BOO. In conclusion, there is a rationale for the selective blockade of TRPV1 channels for various bladder disorders. The AICI model is clinically relevant for the investigation of pathophysiological conditions in which bladder C-fiber afferents are overexcited and for assessing innovative treatments for bladder disorders based on their pathophysiology.

## 1. Introduction

Capsaicin (8-methyl-N-vanillyl-6-nonenamide) is a phenolic compound found in chili peppers that causes a burning sensation in mucous membranes. Other molecules that are structurally and functionally similar to capsaicin include capsaicinoids (dihydrocapsaicin, nordihydrocapsaicin, homodihydrocapsaicin and homocapsaicin), capsinoids (which are less potent) and the extremely potent resiniferoids, the best known of which is resiniferatoxin [1,2]. Capsaicin and resiniferatoxin have been extensively used to increase the understanding of functional lower urinary tract (LUT) disorders and to test in humans for the treatment of various LUT disorders.

The administration of capsaicin to the bladder of neonatal rats destroys a specific subpopulation of sensory nerves and leads to bladder enlargement. This suggests that the sensory nerves that mediate the micturition reflex are, at least in part, capsaicin-sensitive [3]. These nerves trigger neurogenic inflammation [4] and transmit visceral nociception to the CNS. In certain pathological situations, alterations in the function of the sensory nerves reduce the micturition threshold. This reduction can cause overactive bladder/detrusor overactivity (OAB/DO), painful bladder syndrome/interstitial cystitis (PBS/IC), neurogenic detrusor overactivity (NDO) or bladder outlet obstruction (BOO)-related OAB/DO. Capsaicin-sensitive neurons are bipolar, have unmyelinated axons (C-) and their somata are located in dorsal root and trigeminal ganglia. A subset of sensory neurons with thin myelinated axons (Aδ fibers) are also capsaicin sensitive [1].

Capsaicin exerts a biphasic effect on the sensory nerves. Initial excitation is followed by a long-lasting blockade that renders C-fibers resistant to natural stimuli. It has been suggested that these effects result from its action on the vanilloid receptor [5]. The role of capsaicin-sensitive nerves in micturition was demonstrated by bladder instillation of capsaicin in individuals with bladder hypersensitivity disorders, which caused a concentration-related reduction in the first desire to void, bladder capacity and the pressure threshold for micturition, [6]. Those results led to the suggestion that intravesical capsaicin could desensitize sensory nerves and reduce bladder hypersensitivity (6], which was confirmed in a later study that demonstrated that a single intravesical capsaicin instillation reduced neurogenic detrusor overactivity for several months [7].

In 1997, the first vanilloid (capsaicin) receptor was cloned. It was named transient receptor potential vanilloid subfamily, member 1, TRPV1 [8]. This was the starting point for a large number of studies that showed the importance of this receptor for normal bladder function and its role in different types of bladder dysfunction. An in vivo model developed in healthy un-anesthetized rats showed that intravesical capsaicin induces reversible reliable concentration-dependent detrusor hyperactivity assessed by cystometry, which is abolished by intra-arterial hexamethonium administered near the bladder, or by intrathecal morphine [9]. The model was acute and easy to handle and the cystometry results were reliable. Since then, intravesical capsaicin has been widely used in rats to study OAB/DO, painful bladder syndrome/interstitial cystitis (PBS/IC), neurogenic detrusor overactivity (NDO) and bladder outlet obstruction (BOO)-related OAB/DO. It has been postulated that the dysfunction of bladder afferent signals is one of the mechanisms behind these conditions.

This review provides a description of capsaicin and its receptor, TRPV1, its mechanisms of action and its basis for potential clinical use. We also give examples of how the acute intravesical capsaicin rodent model can be applied both to obtain physiological/pathophysiological information that could be translated to the clinical situation and to assess innovative treatments for bladder disorders based on their pathophysiology.

## 2. Materials and Methods

We performed a comprehensive search of the major literature databases (PubMed, Embase and Cochrane) and the abstracts from several conferences (American Urological Association, European Urological association, International Continence Society). We did not limit the search by language (non-English publications were translated) or date. We completed the search in June 2022. The following authors completed the review for the following topics: K-E.A. and F.G. for non-neurogenic overactive bladder/detrusor overactivity, K-E.A. and P.D. for neurogenic detrusor overactivity and painful bladder syndrome/interstitial cystitis, D.B-R. for bladder disorders caused by bladder outlet obstruction.

## 3. Results

### 3.1. TRPV1 and Normal Bladder Function

Capsaicin exerts its actions by binding to the TRPV1 receptor/channel. The expression pattern and properties of the TRPV1 channels in the lower urinary tract have been well described [10,11,12]. TRPV1 is a non-selective cationic channel with high Ca^2+^ permeability that allows the passage of cations, mainly Ca^2+^. It is activated by vanilloids, noxious heat and low pH [8,13]. TRPV1 is the best-characterized member of the TRPV subfamily (TRPV1–6). Its morphology and function in animal models have been well described and several studies have determined the clinical effects of its manipulation [12,14,15]. However, the role of TRPV1 in normal human bladder function has not been fully determined. Cation influx through activated TRPV1 channels induces cell depolarization in afferent nerve fibers. This triggers an action potential, which in turn activates spinal reflexes and/or propagates to the brain to evoke conscious perception of bladder sensations. The depolarization also causes neuropeptide release, which could cause neurogenic inflammation by its action on the receptors of the released agents, e.g., substance P and CGRP.

Most bladder afferents are polymodal and respond to different chemical, thermal and mechanical stimuli, depending on the specific receptor subtypes that they express [16]. Approximately 75% of pelvic nerve fibers are mechanosensitive and respond to bladder stretch [17]. In urothelial cells, the influx of Ca^2+^ through different TRP channels initiates a myriad of Ca^2+^-dependent signaling responses.

A growing body of evidence suggests that, although the majority of afferent nerve fibers are located within the detrusor muscle, urothelial cells also contribute to mechano-sensation and chemo-sensation in the bladder. TRPV1 expression has been found in the suburothelial nerve plexus, detrusor smooth muscle and interstitial cells [12]. Thus, there is evidence of TRPV1 expression in small diameter bladder afferent fibers that lie close to the urothelium and also in bladder sensory neurons within the dorsal root ganglia (DRG).

The role of TRP channels in myogenic activation of afferents is unclear. Several TRP channels have been identified on detrusor muscle cells. TRPV1 channel agonists may have a direct contractile effect on detrusor muscle [18,19,20]. However, the roles of the urothelium and the interstitial cells (ICCs) in afferent activation are complex and have not yet been definitively established [21,22]. The secretion of adenosine triphosphate (ATP) by urothelial bladder cells is an important signal mediator, and suburothelial ICCs respond to purinergic stimulation by firing Ca^2+^ transients [23]. Suburothelial ICCs may be able to affect the activity of the detrusor myocytes [24,25,26].

### 3.2. Intravesical Capsaicin and Role of TRPV1 in LUT Dysfunction

#### 3.2.1. Role of TPRV1 in Non-Neurogenic Overactive Bladder/Detrusor Overactivity

One study showed that TRPV1 expression was significantly higher in women with overactive bladder (OAB) than in women without OAB and that the increased expression was closely correlated to OAB occurrence [27]. Urodynamic variables including maximum flow rate (Qmax), first desire volume, strong desire to void volume, maximum cystometric capacity and bladder compliance were also lower in the individuals with OAB than those without. Likewise, Liu et al., [28] investigated patients with OAB symptoms who had no demonstrable detrusor overactivity (DO) but who had sensory urgency (early first sensation). Furthermore, TRPV1 expression levels in the trigone were inversely correlated with volume at first sensation during bladder filling. In contrast, TRPV1 expression levels were normal in individuals with idiopathic DO (IDO), suggesting sensory urgency and IDO distinct molecular bases [28]. Exposure at an early age to agents that affect TRPV1 channels may predispose to the later development of bladder dysfunction. This was demonstrated in a study in which the bladders of ten-day old rat pups were first desensitized with capsaicin, then sensitized with intravesical acetic acid diluted with saline [29]. Compared with a control group that did not undergo capsaicin desensitization, the acetic acid did not cause significant inflammation; however, it induced bladder sensitization that persisted into adulthood. This finding suggests that TRPV1 activation plays a role in inducing and maintaining bladder sensitization.

Social stress may be a cause of urinary bladder dysfunction in children that could continue into adulthood. Stress can trigger OAB by inducing TRPV1-dependent afferent nerve activity. This was demonstrated in six-week-old male C57BL/6 mice that were exposed to a C57BL/6 retired breeder aggressor mouse (in a barrier cage) [30]. Conscious cystometry was performed with and without intravesical infusion of the TRPV1 inhibitor, capsazepine evidenced that stress reduced in vivo inter-micturition interval and voided volume, which was restored by capsazepine intravesical infusion. Measured pressure–volume relationships and afferent nerve activity during bladder filling using an ex vivo bladder preparation suggested that, at low pressures, bladder compliance and afferent activity were elevated in the mice that were exposed to the stress compared with those that were not. A later study that used an intensified model of social stress demonstrated that TRPV1 channels are also implicated in the development of bladder decompensation and underactivity [31]; this finding suggests that treatments to prevent stress-induced bladder dysfunction in children should aim at TRPV1 receptors.

Evidence suggests that TRPV1-IR (immunoreactive) nerves are present throughout the whole urogenital tract, including the urethra, with capsaicin affecting both urethral smooth and striated muscles [32]. It has been speculated that DO may be initiated from the urethra [33] and, in females, a rapid pattern of urethral pressure variation (“unstable urethra”) seems to be closely associated with DO [34,35,36,37]. This raises the question of whether the TRPV1 channel is involved in urethral functions that can be linked to DO/OAB.
Capsaicin OAB/DO studies

The effect of capsaicin on bladder hyperactivity was evaluated by intravesical instillation in concentrations of 1, 30 and 100 µM in awake rats [9]. Continuous cystometric measurements of micturition pressure (the maximum bladder pressure during micturition), bladder capacity (the volume of saline remaining in the bladder at the latest previous micturition plus the volume of infused saline at micturition), micturition volume (the volume of expelled urine) and residual volume (bladder capacity minus micturition volume) showed that capsaicin induced a concentration-dependent bladder hyperactivity.

Using the same method, a later study confirmed that intravesical capsaicin 30 µM in rats induced DO, increased micturition and basal pressures and decreased bladder capacity and micturition volume [38]. Administration of the nitric oxide synthesis (NOS) inhibitor L-NAME either intrathecally or intra-arterially did not change the effect. Therefore, NO does not appear to be involved in the spinal regulation of the volume- or capsaicin-induced micturition [39]. In contrast, DO induced by capsaicin-mediated C-fiber activation can be abolished by locally released NO [40].

Chronic cyclophosphamide (CYP) administration (intraperitoneal) as an OAB model was used to analyze urodynamic parameters in urethane anesthetized rats [41]. In healthy rats, detrusor contractility was completely inhibited by intravesical instillation of capsaicin (1 mM), thus preventing full voiding. However, lidocaine did not affect micturition cycles. In rats with CYP-induced overactivity, intravesical capsaicin and lidocaine reduced the severity of detrusor overactivity and improved cystometric variables. The authors thus concluded that chronic “chemical” CYP-induced cystitis leads to bladder overactivity in rats and that modulation of C-fiber activity by capsaicin and lidocaine reduces the severity of DO and improves urodynamic parameters. These observations confirm the hypothesis that two types of unmyelinated bladder afferent C-fibers, capsaicin-sensitive and capsaicin-resistant, play a central role in the pathophysiology of overactive bladder.

In a review of the use of resiniferatoxin (RTX), an analog of capsaicin, to treat IDO, Cruz and Dinis [42] found that RTX increased the volume to first detrusor contraction and bladder capacity, reduced urinary incontinence and improved quality of life. RTX might also be effective for the treatment of urgency, which is the primary symptom of OAB.
Capsaicin for the assessment of innovative treatments for OAB/DO

JTS-653 is a selective TRPV1 antagonist. Administration of this compound five minutes after intravesical infusion of 100 µM capsaicin in anesthetized rats dependently inhibited the increase in pelvic nerve discharge and intravesical pressure that had been elicited by the capsaicin [43]. Although the clinical development of this compound for the treatment of OAB stopped in 2010, the results of that study provide evidence of the involvement of TRPV1 in bladder overactivity via afferent nerve activation elicited by capsaicin.

A non-peptidergic competitive antagonist of NK2 receptors, SR 48,968, considerably reduced or even abolished bladder hyperactivity elicited by 30 µM capsaicin in conscious rats [9]. Another NK2 receptor antagonist, Nepadutant (MEN 11,420 (100 nmol/kg, i.v.)), reduced the number of reflex contractions and the area under the curve of bladder pressure after intravesical administration of capsaicin (6 nmol/0.6 mL) in anesthetized rats. These in vivo results show that NK2 receptor antagonists reduce most of the effects of intravesical capsaicin in the rat urinary bladder. Therefore, the stimulation of NK2 receptors by tachykinins, which are released from capsaicin sensitive nerves, may lower the threshold for the initiation of the micturition reflex [44]. The authors concluded that, since NK2 receptor antagonists do not modify the micturition reflex evoked by bladder distension, this class of drugs could be a safe treatment for bladder hyperreflexia. However, although several patent applications have been filed for the use of NK2 antagonists to treat overactive bladder [45], no clinical studies have yet assessed these compounds. Moreover, at the time of writing, no preclinical evaluations have begun to evaluate the treatment of OAB by NK2 antagonists [46].

NS4591, a positive modulator of calcium-activated potassium channels of intermediate (IK; KCa3) and small (SK; KCa2) conductance, inhibited bladder overactivity in rats induced by capsaicin [47]. Its effects on bladder overactivity were evaluated in conscious unrestrained female rats, some of which received this compound as a pretreatment, either orally or intraperitoneally, before the administration of capsaicin (6 µM). After a 60 min baseline recording, capsaicin was administered for around 20 min, then saline was administered for 60 min to induce bladder overactivity. In the rats that did not receive the pretreatment, micturition volume and inter-contraction interval (ICI) decreased and maximal micturition pressure increased. This response did not occur in the rats that received NS4591. This compound has also been found to inhibit the firing of action potentials in bladder afferent neurons in rats; however, it has not yet been tested in humans.

Acute intravesical capsaicin has also been used to study the effects of intradetrusor gene therapy vector delivery on bladder hyperactivity in rats. To assess the effect of intravesical injection of replication-defective recombinant herpes simplex virus type 1 (HSV-1) vectors encoding preproenkephalin and containing the human cytomegalovirus promoter, capsaicin 15 µM in 10% ethanol 10% Tween 80 and 80% saline was instilled into the bladder at 0.04 mL/min. The reduction in the inter-contraction interval elicited by capsaicin was significantly decreased by the vectors one week after intravesical injection. The vectors were transported through bladder afferent pathways to the dorsal root ganglia (DRG), which is where bladder afferent nerves originate [48].
Conclusion

Altogether, these data suggest that acute intravesical capsaicin is a suitable model for the study of afferent hyperactivity caused by stimuli that trigger peptide release from sensory nerves in the bladder and mimic human OAB syndrome.

#### 3.2.2. Role of TRPV1 in Neurogenic Detrusor Overactivity

Overactive bladder and voiding dysfunction are the most common symptoms of neurogenic lower urinary tract dysfunction [49,50]. The neurological dysfunction causes an inability to suppress spontaneous detrusor contractions. Neurogenic detrusor overactivity (NDO) may result from suprapontine or spinal cord lesions (above the lumbosacral level). Voluntary control of micturition is abolished by lesions situated above the lumbosacral cord level. This causes initial bladder areflexia and complete urinary retention but is followed by the slow development of a sacral spinal reflex mediated by formerly silent capsaicin-sensitive unmyelinated C-fibers. This reflex is triggered by low-volume bladder filling and causes NDO and detrusor external sphincter dyssynergia (DESD).

The emergence of bladder C-fiber reflexes may be mediated by neurotrophic factors, such as nerve growth factor (NGF) and brain derived neurotrophic factor (BDNF). In rats, spinal cord injury (SCI) increases NGF production in the bladder, spinal cord and dorsal root ganglia. BDNF represses the growth of sensory afferents during the initial phase of spinal shock, thus protecting from bladder overactivity; however, once NDO has developed, BDNF appears to maintain the condition. Urothelium/suburothelium sensitization of afferent nerve fibers plays a role in the pathophysiology of NDO. Individuals with NDO have increased numbers of suburothelial nerve fibers that are immunoreactive to P2X3 compared with healthy controls.

Evidence suggests that NDO is mediated by capsaicin-sensitive C-fiber afferents, and the role of TRPV1 in the pathophysiology and treatment of neurogenic detrusor overactivity (NDO) has been well demonstrated [51,52,53]. Studies of individuals with NDO found increased immunoreactivity of PGP9.5 (nerve stain) and TRPV1 in the suburothelium and basal layers of the urothelium compared with controls. TRPV1 immunoreactivity was significantly lower in individuals with NDO who responded clinically to intravesical resiniferatoxin (RTX), which suggests that TRPV1 is involved in the pathophysiology of NDO [51,52,53]. However, the effects of vanilloids (capsaicin, RTX) on urothelial TRPV1 indicates that vanilloid actions are more complex than simple C-fiber desensitization.

In people with NDO, the sacral micturition reflex is inactivated by C-fiber desensitization [54]. This sacral reflex emerges after chronic spinal cord lesions above the sacral level of the sacral level [54,55,56]. Neuronal and urothelial TRPV1 expression is increased in the bladder in people with NDO [51,57], and the degree of expression is correlated with urgency sensation [28]. Intravesical application of vanilloids lowers TRPV1 expression [52,53].
Capsaicin NDO studies

Several non-controlled studies have evaluated the effect of capsaicin 1–2 mM, 30% alcohol solution instilled in the bladder on NDO in humans [7,56,58,59,60,61]. The best clinical outcomes were found in individuals with incomplete spinal cord lesions. In the only controlled study [62], incontinence and urge sensation reduced in all ten individuals who received capsaicin, whereas improvement only occurred in one of the ten who received the alcohol solution. Despite these encouraging results, the use of alcoholic capsaicin solutions is limited by their potency and side effects and we found no recent studies using such solutions.
Capsaicin for the assessment of NDO treatments

Intradetrusor botulinum toxin A (BTX/A) is registered as second line treatment for NDO in individuals who are resistant or intolerant to oral antimuscarinics. BTX/A inhibits neurotransmission between post-ganglionic parasympathetic nerve terminals and detrusor muscle fibers and also targets bladder afferents. BTX/A impairs bladder afferent firing elicited by acute intravesical capsaicin instillation in chronic spinal cord transected mice [63]. In chronic thoracic spinal cord transected rats, non-voiding contractions during the filling phase generated by the recruitment of afferent C-s are significantly reduced by BTX/A [64]. BTX/A decreases contractile responses in in vitro rat bladder strips to capsaicin [65]. BTX/A significantly reduced CGRP release induced by capsaicin in isolated rat bladder preparations [66].

Nerve growth factor antibodies (NGF-Ab) have been studied for the treatment of chronic pain. NGF overexpression plays a major role in the hyperexcitability of capsaicin-sensitive bladder C-fibers, resulting in NDO after spinal cord injury. In chronic SCI mice, the change in electrophysiological properties of C-fibers was reversed by NGF-Ab. In these experiments, bladder C-fibers were identified with whole patch clamp recordings of L6-S1 afferents sensitive to capsaicin [67].
Conclusion

Although not essential for the initiation of reflex micturition in SCI rats, capsaicin-sensitive bladder afferents contribute to bladder hyperactivity during the filling phase. Therefore, studies should seek to develop and evaluate pharmaceutical products that target capsaicin-sensitive bladder afferents for the treatment of bladder hyperreflexia after SCI [68]. Neurotrophic factors released after SCI induce plasticity of these afferents and this plays a crucial role in the pathophysiology of NDO. This led to the clinical evaluation of intravesical capsaicin for the treatment of NDO, opening new avenues for innovative treatments targeting bladder afferents to control NDO incontinence.

#### 3.2.3. Role of TRPV1 in Painful Bladder Syndrome/Interstitial Cystitis (PBS/IC)

Painful bladder syndrome/interstitial cystitis (PBS/IC) causes persistent pelvic pain and lower urinary tract symptoms that reduce the quality of life of affected individuals. The phenotypes and etiologies of this condition vary widely. 

No predictive animal models of PBS/IC currently exist [69]. Cyclophosphamide-induced bladder overactivity has been widely used but does not reflect all the characteristics of the disorder. Evidence from pathological and genomic studies suggests that PBS/IC should be categorized according to whether Hunner lesions are present (IC) or not (PBS) and not by clinical phenotyping according to the symptoms experienced [70]. IC may constitute a separate entity. Studies have found TRPV1 immunoreactivity to be altered in individuals with PBS/IC, which suggests that TRPV1 channels are involved in the pathogenesis of PBS/IC. Bladder biopsies from individuals with PBS who met the National Institute for Diabetes and Digestive and Kidney Diseases (NIDDK) research criteria for IC showed increased TRPV1 immunoreactivity within the nerve fibers compared with biopsies from individuals with asymptomatic microscopic hematuria [71]. Furthermore, pain ratings were correlated with the relative density of TRPV1 nerve fibers. Another study also involving bladder biopsies from individuals with IC/BPS who met the NIDDK research criteria found increased severity of inflammation that was correlated with a higher TRPV1 immunoreactive nerve fiber density and higher NGF levels [72] compared with control participants. Suburothelial TRPV1-immunoreactive nerve fiber density was significantly correlated with pain scores and urgency scores. The density of PGP9.5-immunoreactive nerve fibers was significantly increased in those with IC/BPS and was positively correlated with inflammation severity. Thus, higher levels of expression of TRPV1-immunoreactive nerve fibers and NGF cause more severe inflammation and clinical symptoms in IC/BPS.
Capsaicin PBS/IC studies

During a study using intravesical capsaicin in rats, nociceptive behavior was observed. The behavior was considerably reduced by intrathecal, and partially reduced by intra-arterial, administration of L-NAME [38]. However, L-NAME administration had no effect on cystometric variables, which suggested that the capsaicin-induced nociceptive behavior mainly involves spinal NO. This raised the question as to whether NO can be used to counteract capsaicin-induced pain, which has been a limiting factor for its intravesical use in individuals with OAB/DO. This question was partly addressed in a study of bladder activity during intravesical capsaicin (30 µM) instillation using continuous infusion cystometry in urethane-anesthetized rats [39]. The results showed that locally released NO suppressed DO induced by capsaicin-mediated C-fiber activation and confirmed that central NO pathways are not involved in capsaicin-induced DO. A later study confirmed the involvement of the NO/cGMP signaling pathway, with a predominant action on the sensory rather than on the motor component of the micturition reflex, in a rat model of capsaicin-induced bladder overactivity associated with C-fiber afferent activation [40]. The effects of GRC-6211, an orally active TRPV1 antagonist, on the function and noxious input of naïve and inflamed bladders were analysed in urethane anesthetized rats [73]. Cystometry during infusion of saline, 100 µM capsaicin or 0.5% acetic acid showed that GRC-6211 (0.1 mg/kg) completely prevented capsaicin induced irritation, which suggests that TRPV1 antagonists might be useful for the treatment of cystitis. Preproenkephalin, a gene encoding for endogenous opioids, has been expressed in bladder afferents by gene transfer using replication-defective herpes simplex virus type 1 (HSV-1) vectors in a rat model of bladder hyperactivity and pain [48]. Capsaicin (15 µM in 10% ethanol 10% Tween 80 and 80% saline) was intravesically instilled at 0.04 mL/min. Intravesical vector injections decreased bladder hyperactivity and pain behaviors induced by intravesical application of capsaicin

The evidence for the involvement of TRPV1 in pain perception led several groups to evaluate RTX as a treatment for bladder pain in IC/PBS [74,75,76,77,78]. However, a randomized double-blind study in 163 individuals with IC/PBS found no significant effect of RTX (10 nmol/L, 50 nmol/L and 100 nmol/L) in terms of overall symptoms, pain, urgency, frequency, nocturia or average voided volume during 12 weeks of follow-up compared with a placebo treatment [79].
Capsaicin for the assessment of innovative treatments for IC/PBS

GRP 18, a member of the cannabinoid receptors family, is involved in the regulation of pain and bladder function by inhibition of TRPV1. In an intravesical cyclophosphamide rat model of IC/BPS, intrathecal injection of a GRP18 agonist, resolvin D2, increased the micturition interval during cystometry. In cyclophosphamide treated rats, resolvin D2 also inhibited calcium influx induced by capsaicin 1 µM in a primary culture of L6-S1 dorsal root ganglia in which most of the cell bodies of bladder afferent neurons are located. [80]. The nociceptin/orphanin FQ peptide receptor (NOP), activated by its endogenous peptide ligand nociceptin/orphanin FQ (N/OFQ) has several effects, including modulation of pain signaling. The density of NOP-positive nerve fibers within the bladder suburothelium were found to be significantly increased in individuals with detrusor overactivity and IC/PBS. In cultured human and rat dorsal root ganglia (DRG) neurons, N/OFQ decreased calcium influx in response to 200 nM to 1 µM capsaicin stimuli, thus supporting further investigations into N/OFQ instillation for the treatment of IC/PBS [81]. Gene therapy using non-replicant defective Herpes virus vectors expressing pre-proenkephalin gene injected into the rat bladder wall decreased bladder hyperactivity after intravesical instillation of capsaicin (15 µmol/L). This effect was reversed by naloxone, thus confirming the involvement of enkephalinergic mechanisms to control bladder pain [48,82].
Conclusion

Current therapies for IC/PBS have limited clinical effects on this frustrating and debilitating disease. New therapies are therefore required for those with refractory IC/BPS. TRPV1 channels seem to be involved in some IC/PBS phenotypes. Experimental research and the development of drugs that target bladder afferents are needed. The acute intra-bladder capsaicin instillation model appears relevant for the preclinical assessment of such treatments.

#### 3.2.4. Role of TRPV1 in Bladder Disorders Caused by Bladder Outlet Obstruction

Comparison of the involvement of tachykinins via NK1 receptors in the micturition reflex induced by bladder filling in normal rats and in rats with bladder hypertrophy due to BOO has shown that the increase in the afferent input from the bladder to the dorsal root ganglion during bladder filling is at least partly conveyed by capsaicin-sensitive afferents and even more so in the BOO rat [83]. The ice water test (IWT) was developed for clinical practice to identify C-fiber involvement in the micturition reflex by detecting uninhibited detrusor contractions after instillation of iced water into the bladder. The response rate is considerably higher in individuals with neurological disease than those without (around 70% versus 27%). However, among responders with no known neurological disease, the degree of BOO was found to be significantly greater than in non-responders [84]. Positive responses to the IWT have also been reported in individuals with BPH/BOO [85]: 71% of individuals with BOO responded compared with 7% in those with no BOO.
Studies of the effect of capsaicin on bladder dysfunction caused by BOO

Clinical studies reported that phosphodiesterase type 5 inhibitors (PDE5-Is) not only improved LUT voiding symptoms (LUTS) that were suggestive of BPH, but also storage symptoms. Furthermore, the effect was more pronounced on storage than on voiding symptoms [86,87,88]. To facilitate understanding of these results, the effects of the key components of the NO/cGMP signaling pathway were studied in a model of bladder hyperactivity resulting from C-fiber activation by capsaicin. The results showed that NO exerted an inhibitory effect on the micturition reflex via the soluble guanylate cyclase (sGC)/cGMP pathway, since 8Br-cGMP and PDE5-I mimicked the inhibitory effect of the NO donor sodium nitroprusside, whereas the inhibition of nitric oxide synthase by L-NAME and sGC by LY-83583 had the opposite effect [40]. These findings corroborate the benefit observed with PDE5-Is on storage LUTS that is suggestive of BPH in men [86,87,88].

We were unable to find any reports of the use of acute capsaicin in a BOO model to assess the role of capsaicin-sensitive afferents in outflow obstruction for the testing of innovative treatments.
Conclusion

The development of new effective drugs for the treatment of bladder disorders due to BOO is essential. The use of a versatile experimental model such as the capsaicin-induced hyperactivity bladder model appears relevant, particularly for the study of bladder disorders caused by C-fiber hyperexcitability.

### 3.3. Clinical Application of Drugs Blocking TRPV1 Receptors

Despite capsaicin and resiniferatoxin blocking TRPV1 receptors by causing receptor desensitization seeming to have efficacy in the treatment of neurogenic bladder, they are currently rarely or no longer used clinically [89]. As mentioned previously, based on results from several animal models of LUT disorders, selective TRPV1 antagonists appear to be promising drug candidates. However, polymodal first generation selective TRPV1 antagonists cause hyperthermia, both in animal models and in humans [90,91], which limits their clinical application. Second-generation (mode-selective) TRPV1 antagonists potently block channel activation by capsaicin, but exert different effects (e.g., potentiation, no effect, or low-potency inhibition) in the proton mode, heat mode or both [91]. Brown et al., [92] conducted a phase I study in healthy volunteers on the safety and pharmacokinetics of oral NEO6860, a modality selective TRPV1 antagonist, and found no clinically significant increase in temperature or heat pain threshold/tolerance, but a significant antagonistic effect on intra-dermal capsaicin-induced pain. A large number of TRPV1 antagonists has been tested in phase I studies [91], but few have been further developed [93] and none has so far been approved [91,92,93]. There seems to be no published proof-of-concept studies on LUT disorders.

## 4. General Conclusions (Table 1)

TRPV1 channels are highly expressed in C-fibers that innervate the urinary bladder and urethra and have been shown to play an integral role in modulating the excitability of bladder afferents and in hypersensitivity induced by bladder inflammation. This review was not systematic, which does represent a limitation. Despite this drawback, it appears that elective TRPV1 channel antagonists have shown promising effects in several animal models of LUT dysfunction, which makes them potential drug targets. A drawback with the small molecule TRPV1 antagonists produced so far is that acute pharmacological inhibition of TRPV1 evokes hyperthermia, independently of the chemical structure of the antagonist [90]. However, since there is a strong rationale for the blockade of TRP channels, further research is justified.

In our opinion, the acute intravesical capsaicin rodent model can be applied both to obtain further physiological/pathophysiological information that can be translated to the clinical situation and to assess innovative treatments for bladder disorders based on their pathophysiology.

**Table 1 medsci-10-00050-t001:** Summary of the data regarding (i) the involvement of TRPV1 sensory fibers in the pathophysiology of lower urinary tract dysfunctions (LUTD), (ii) the relevance of acute intravesical capsaicin instillation (AICI) in rodents to mimic the condition, (iii) clinical trials assessing the effect of vanilloids for the treatment of the condition and (iv) main outcomes of the clinical trials. RTX—resiniferatoxin, SCI—spinal cord injury.

LUTD	Involvement of TRPV1 Receptors in LUTD	AICI in Rodents	Clinical Trials Assessing the Effect of Vanilloids	Clinical Efficiency of Vanilloids
OAB/DO	Yes	Yes	Yes	RTX decreased incontinence episodes and improved quality of life
NDO	Yes	Yes	Yes	Capsaicin decreased incontinence episodes in incomplete SCI
BOO	Yes	Yes	No	-
IC/PBS	Yes	Yes	Yes	RTX did not improve symptoms

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
