# Peer review of "Acute Intravesical Capsaicin for the Study of TRPV1 in the Lower Urinary Tract: Clinical Relevance and Potential for Innovation"

_medsci, 2022, doi:10.3390/medsci10030050_

Round 1

Reviewer 1 Report

Andersson et al. they present an interesting manuscript well structured and with interesting data. The authors should review minor points such as being able to include a specific section of applicability in the clinic. The authors must include a section on clinical trials and updated treatments.

Finally, I would ask the authors to include an illustrative figure of the main conclusions together with a table.

Author Response

Response to Comments and Suggestions for Authors by Reviewer 1 :

We highly appreciated the constructive criticism you made of our manuscript. We have tried our best to improve the manuscript in accordance with your comments and suggestions. We hope that these changes will meet your approval. Thank you again for your thorough review of our work.

Andersson et al. they present an interesting manuscript well structured and with interesting data. The authors should review minor points such as being able to include a specific section of applicability in the clinic. The authors must include a section on clinical trials and updated treatments.

Thank you for this suggestion.

We have added a section with a few references prior to the conclusion regarding applicability in the clinic, clinical trials and updated treaments (lines 377-390).

Finally, I would ask the authors to include an illustrative figure of the main conclusions together with a table.

We have added a table (pp 14-15) summarizing the main conclusions.

Reviewer 2 Report

I believe it’s an useful review and well written manuscript. Here some advice:

-          In methods “We performed a comprehensive search of all major literature databases and the abstracts from several conferences”. Please better define the databases screened. Please also define at least the names of the major conferences.

-          Who completed the abstract/articles screening (initials)? I believe it is an info that should be inserted even if this isn’t a systematic review.

-          Line 238. The title “conclusion” is missed.

-          However, I actually would remove the “conclusion” from every single sub-chapter. I actually believe that the authors should write a wider conclusion (instead of “general conclusions”) at the end of the manuscript, with inputs and considerations from all the items evaluated in the sub-chapters.

-          Limitations should be declared at the end of the manuscript. These can be inserted in the conclusions.

Author Response

Response to Comments and Suggestions for Authors by Reviewer 2 :

We highly appreciated the constructive criticism you made of our manuscript. We have tried our best to improve the manuscript in accordance with your comments and suggestions. We hope that these changes will meet your approval. Thank you again for your thorough review of our work.

I believe it’s an useful review and well written manuscript.

Here some advice:

-          In methods “We performed a comprehensive search of all major literature databases and the abstracts from several conferences”. Please better define the databases screened. Please also define at least the names of the major conferences.

These information have been added in the revised manuscript (lines 98-99).

-          Who completed the abstract/articles screening (initials)? I believe it is an info that should be inserted even if this isn’t a systematic review.

      This information has also been added in the revised version (lines 101-105).

-          Line 238. The title “conclusion” is missed.

      corrected

-          However, I actually would remove the “conclusion” from every single sub-chapter. I actually believe that the authors should write a wider conclusion (instead of “general conclusions”) at the end of the manuscript, with inputs and considerations from all the items evaluated in the sub-chapters.

      In order to take into account suggestions form reviewers 1 and 2,  we would rather propose to add to the general conclusion a table ( pp14-15) summarizing the inputs and considerations from all the items evaluated in the sub-chapters. We hope that this will be acceptable for this reviewer.

-          Limitations should be declared at the end of the manuscript. These can be inserted in the conclusions.

      Thank you this suggestion. Limitations have been added in the conclusion (line 394).

Round 2

Reviewer 2 Report

I agree with editings made by the authors